# Rift Valley Fever Virus Exposure amongst Farmers, Farm Workers, and Veterinary Professionals in Central South Africa

**DOI:** 10.3390/v11020140

**Published:** 2019-02-07

**Authors:** Veerle Msimang, Peter N. Thompson, Petrus Jansen van Vuren, Stefano Tempia, Claudia Cordel, Joe Kgaladi, Jimmy Khosa, Felicity J. Burt, Janice Liang, Melinda K. Rostal, William B. Karesh, Janusz T. Paweska

**Affiliations:** 1Epidemiology Section, Department of Animal Production Studies; Faculty of Veterinary Science, University of Pretoria, Onderstepoort 0110, South Africa; 2Centre for Emerging Zoonotic and Parasitic Diseases, National Institute for Communicable Diseases, National Health Laboratory Service, Sandringham 2192, South Africa; petrusv@nicd.ac.za (P.J.v.V.); joek@nicd.ac.za (J.K.); 3MassGenics, Duluth, GA 30026, USA; stefanot@nicd.ac.za; 4Influenza Division, Centers for Disease Control and Prevention, Pretoria 0001, South Africa; 5Influenza Division and Centers for Disease Control and Prevention, Atlanta, GA 30301, USA; 6ExecuVet (Pty) LTD, Bloemfontein 9300, South Africa; execuvet26@gmail.com; 7National Institute for Communicable Diseases, National Health Laboratory Service, Sandringham 2192, South Africa; jimmyk@nicd.ac.za; 8Division of Virology, National Health Laboratory Service and Faculty of Health Sciences, University of the Free State, Bloemfontein 9300, South Africa; BurtFJ@ufs.ac.za; 9EcoHealth Alliance, New York, NY 10001, USA; janice.en.liang@gmail.com (J.L.); rostal@ecohealthalliance.org (M.K.R.); karesh@ecohealthalliance.org (W.B.K.)

**Keywords:** Rift Valley fever virus, emerging disease, South Africa, seroprevalence, human exposure, statistical case estimation, spatial distribution

## Abstract

Rift Valley fever (RVF) is a re-emerging arboviral disease of public health and veterinary importance in Africa and the Arabian Peninsula. Major RVF epidemics were documented in South Africa in 1950–1951, 1974–1975, and 2010–2011. The number of individuals infected during these outbreaks has, however, not been accurately estimated. A total of 823 people in close occupational contact with livestock were interviewed and sampled over a six-month period in 2015–2016 within a 40,000 km^2^ study area encompassing parts of the Free State and Northern Cape provinces that were affected during the 2010–2011 outbreak. Seroprevalence of RVF virus (RVFV) was 9.1% (95% Confidence Interval (CI95%): 7.2–11.5%) in people working or residing on livestock or game farms and 8.0% in veterinary professionals. The highest seroprevalence (SP = 15.4%; CI95%: 11.4–20.3%) was detected in older age groups (≥40 years old) that had experienced more than one known large epidemic compared to the younger participants (SP = 4.3%; CI95%: 2.6–7.3%). The highest seroprevalence was in addition found in people who injected animals, collected blood samples (Odds ratio (OR) = 2.3; CI95%: 1.0–5.3), slaughtered animals (OR = 3.9; CI95%: 1.2–12.9) and consumed meat from an animal found dead (OR = 3.1; CI95%: 1.5–6.6), or worked on farms with dams for water storage (OR = 2.7; CI95%: 1.0–6.9). We estimated the number of historical RVFV infections of farm staff in the study area to be most likely 3849 and 95% credible interval between 2635 and 5374 based on seroprevalence of 9.1% and national census data. We conclude that human RVF cases were highly underdiagnosed and heterogeneously distributed. Improving precautions during injection, sample collection, slaughtering, and meat processing for consumption, and using personal protective equipment during outbreaks, could lower the risk of RVFV infection.

## 1. Introduction

Rift Valley fever (RVF) is an important emerging, zoonotic, mosquito-borne disease that causes periodic outbreaks in ruminants and febrile illness in humans [1,2]. The disease is caused by an RNA virus which belongs to the genus *Phlebovirus* within the *Phenuiviridae* family, order of Bunyavirales [3]. Approximately 80–90% of individuals infected with RVF virus (RVFV) manifest symptoms of influenza-like illness [4], with a reported overall case-fatality rate of 1–3%, but as high as 50% among patients with hemorrhagic fever, hepatitis, and renal failure [5]. Retinitis occurs in up to 2% of RVF cases [6,7]. During RVF outbreaks, infection in livestock leads to increased occupational risk for humans exposed to tissues and fluids of infected animals [8]. Individuals at increased risk of RVFV infection include farmers and farm workers, veterinary professionals and those employed in the animal processing industry [9,10]. Humans in these professions often serve as sentinels of RVFV outbreaks even though the disease usually occurs first in animals and then in humans [11]. Inhalation of aerosols during slaughter of infected animals or inoculation via needle-stick or injury or broken skin are other routes of transmission in aforementioned occupational groups [7]. Laboratory-acquired RVFV infections have also been reported [12,13]. General population may become susceptible to RVFV infection by consuming raw milk or via mosquito bites, but no human-to human transmission [7] has been documented.

The first RVF outbreak documented in South Africa occurred in 1950–1951 on the interior plateau (Free State, Eastern Cape and Northern Cape Provinces) [14,15], followed by a second major outbreak in 1974–1975 [16]. The most recent major outbreaks in South Africa occurred during 2010–2011 [17]. After this outbreak, there were no RVF human or animal cases confirmed in South Africa until May 2018, when an isolated outbreak was detected on a single farm in western Free State Province [18,19]. The central plateau of South Africa is a RVF outbreak-prone area where more frequent and intensive outbreaks have occurred compared to the eastern coastal area [20]. 

In South Africa, little is known about the seroprevalence and associated risk factors of human RVFV exposure in the farm environment [16,17]. This study aimed to estimate the seroprevalence of RVFV and to identify hotspots of exposure and factors associated with RVFV infection amongst farmers, farm workers, and veterinary personnel in an epidemic-prone area in South Africa (the central plateau) four years after the 2010–2011 outbreaks. A better identification of these factors will aid in improvement of targeted prevention measures. Further, we aimed to estimate the number of human RVFV infections that had occurred in the farm population in the study area during the previous outbreaks. This study was conducted within a one-health framework for the investigation of the epidemiology of RVF in South Africa.

## 2. Materials and Methods

### 2.1. Ethics Statement

This project was conducted under the protocol approved by the US Hummingbird Institutional Review Board (no. 2014–25 24/11/2014), US DTRA Research Oversight Board (CT 2014–33 27/01/2015), SA Witwatersrand and Pretoria Universities Human Ethics Committee (M140306 30/04/2014; 140/2018 11/06/2018), and SA Provincial Departments of Health Free State and Northern Cape (NC2015/001 09/02/2015; 04/04/2015). Voluntary written consent was obtained from all participants included in the study.

### 2.2. Study Design and Data Collection

We conducted a cross-sectional serological survey during October 2015–February 2016 using single stage cluster sampling of healthy participants aged >11 years in a 40,000 km^2^ area situated between Bloemfontein (Latitude: –29.081885; Longitude: 26.162902 and Mokala National Park (Latitude: –29.132526; Longitude: 24.322333) in the Free State and Northern Cape Provinces. This area experienced a high number of RVFV infections in livestock during 2010–2011. For the survey we targeted individuals at high risk for RVFV infection, including livestock and game farmers and farm workers, and members of livestock (cattle, sheep, or goats)-owning households (livestock farm population; LSFP); and para-veterinary workers and veterinarians (animal health-care workers; AHCW). We estimated a sample size of 770 individuals for an expected RVFV seroprevalence of 50% with 95% confidence intervals, 5% precision, 0.2 intra-cluster correlation, and an average cluster (farm) size of 6.

For the selection of farms, since no complete list of farms was available, we generated random geographic coordinates within the study area using ArcGIS 10.2 (Esri, Redlands, CA, USA) and projected them on Google Earth (Google LLC, Mountain View, CA, USA). The coordinates were selected with probability proportional to the density of livestock-owning households calculated using the 2011 National Census data [21]. The closest farm to each random coordinate was invited to participate to the study. If the farm owner refused participation, we invited the next closest farm willing to participate. All consenting target individuals in the selected farms were invited to participate. For the AHCW group we obtained a complete sampling frame of veterinary professionals working within the study area from the South African Veterinary Council register.

We collected data on demographics, RVF-related knowledge, attitudes, and practices and work-, exposure-, and health-related information using a pre-tested, standardized questionnaire. A farm-level survey was also conducted to obtain farm characteristics. English, Afrikaans, and Sesotho versions of the questionnaires were loaded as an Open Data Kit application [22] on tablets for self-administered use or researcher-assisted interview.

### 2.3. Sample Collection and Laboratory Procedures

Blood was collected by venipuncture into two 8.5 mL serum separator tubes by a South African Nursing Council–registered nurse. Following centrifugation, serum was refrigerated (4 °C) until delivery to the National Institute for Communicable Diseases (NICD) for testing at the Arbovirus Reference Laboratory of the Centre for Emerging Zoonotic and Parasitic Diseases (CEZPD) (Johannesburg, South Africa). 

Human sera were screened for IgG antibodies using an indirect ELISA based on a recombinant nucleocapsid antigen of RVFV using a cut-off value of 28.9 percentage positivity relative to a positive control (sensitivity, *Se* = 99.72%; specificity, *Sp* = 99.62%), as previously described [23]. Positive samples were confirmed using a cut-off value of 38.6 percentage inhibition relative to a positive control, by inhibition ELISA based on whole RVFV antigen (*Se* = 99.47%; *Sp* = 99.66%), as previously described [24].

### 2.4. Statistical Analyses

We estimated proportion of seroprevalence and constructed confidence intervals adjusted for clustering at farm level by using the linearized variance estimator based on a first-order Taylor series linear approximation [25]. In addition, we adjusted the apparent seroprevalence by test sensitivity and specificity as follows: *TP = (AP + Sp − 1)/(Se + Sp − 1)* where *TP* = true prevalence, *AP* = apparent prevalence, *Sp* = specificity, and *Se* = sensitivity. We used the sensitivity and specificity of the two tests in series assuming conditional independence: *Se = Se1 × Se2* and *Sp = 1 − (1 − Sp1) × (1 − Sp2)* [26,27].

We used logistic regression to assess the association between potential risk factors and the apparent RVFV serological status of study participants. For the multivariable logistic regression model, we included all variables with *p* < 0.2 on univariable analysis and then dropped non-significant factors (*p* ≥ 0.05) with manual backward elimination. Some variables (e.g., job description on farm and the ownership of the land used by the farmer (private or communal)) were included in the model regardless of their significance in order to control for confounding. Analysis was done using Stata 13 (StataCorp, College Station, TX, USA) with adjustment for data that were collected using a survey sampling design and clustering using the svy-set command that specified the farm identifier as the primary sampling unit (cluster) variable and svy-prefix for estimation and risk factor analysis commands.

The historical cumulative number of RVF cases that occurred in the study area was estimated as the product of the true seroprevalence, the number of livestock-owning households (LOHH) in the study area, and the number of employees per LOHH in the study area (2015–2016). A probability distribution of RVF case numbers with median and 95% credible interval (2.5th and 97.5th percentiles of distribution) was obtained via 10,000 Monte Carlo simulations using @Risk (Palisade Corporation, Ithaca, NY, USA). Each simulation sampled from the following probability distributions for the input factors:*Seroprevalence ~ Beta(α, β)*
*No. of LOHH ~ Pert(a, b, c)*
*No. of employees per LOHH ~ Normal(μ, σ)*
where *α* = total number of RVFV-seropositive farm workers, *β* = total number of people sampled – α, *a* = minimum no. of LOHH, *b* = most likely no. of LOHH, *c* = maximum no. of LOHH estimates, μ = mean number of employees on the farm, σ = standard error for μ. LOHH estimates were made using data available from the 2011 National Census ((K. Parry Statistics South Africa 2014, pers. Comm.) as described below. Data were based on 1541 “small areas,” a census-defined geographic area, within the study area. The numbers of LOHH per small area were obtained separately for cattle, for sheep and for goats and it was not possible to determine the total number of households that owned cattle or sheep or goats. The high LOHH estimate (*c*) assumed that no farms kept more than one species, the low LOHH estimate (*a*) assumed that every household kept all three species and the most likely LOHH estimate (*b*) assumed most likely in-between scenario calculated as average of *a* and *c*.

The geographical coordinates of the farms where participants that had worked/lived at that farm for four years or longer (therefore were there during the 2010–2011 outbreaks) were sampled-were used for mapping the spatial distribution of the proportion of people that were seropositive on each farm and these data were used for hotspot analysis using ArcGIS 10.5. (Esri, Redlands, California, USA). We first checked overall pattern of the farm seroprevalence data by measuring spatial autocorrelation (Moran’s *I* statistic), which was transformed to a *z*-score, a measure of standard deviation from the mean from a normal distribution in which values greater than 1.96 or smaller than −1.96 indicate spatial autocorrelation that is significant at the 5% level [28]. At the same time, we examined consistency of the spatial pattern of the variable across the study area to evaluate effectiveness of global Moran’s *I* tool. Next, we calculated the Getis-Ord Gi* statistic (*z*-score) by comparing the local situation, i.e., sum of the value for a farm in question and those of neighboring farms to the global situation, i.e., sum of all farm values [29]. 

This identifies local clusters (of farms) that have higher seropositivity values than expected by random chance (hotspots). We used Inverse Distance Weighted (IDW) as interpolation method that averaged the *z*-scores of the measured farm points to predict the *z*-score for the location points without measurement in the interpolated raster cell surface to create the hotspot layer. We also predicted mean RVFV seroprevalence of people that had worked/lived at that farm for four years or longer by fitting a logistic regression model to age data in Stata. We subtracted predicted from actual mean farm seroprevalence of human RVFV and ran a hotspot analysis on the difference in seroprevalences. This was done to verify whether external factors other than age caused hot and cold spots. 

## 3. Results

### 3.1. Study Population

A total of 823 individuals were enrolled and tested during October 2015–February 2016. The median age of participants with available data was 36 years (range 16–84 years). There was a much higher number male (93% (634/684)) than female (7% (50/684)) LSFPs encountered and sampled on farms and amongst AHCWs sex was more equally distributed between men (49% (68/138) and women (51% (70/138)). Most, 685 (83%) were LSFPs from 204 farms and 138 were AHCWs. Of the LSFPs 669 (98%) were from 199 domestic animal farms and the remaining 16 were from five farms that primarily farmed game. The majority, 641 (94%) participants were from 185 privately-owned farms and 44 participants were from 19 communal land farms. The median size of privately-owned farms was 1001–2000 ha. Six percent of farm owners refused participation, but these were replaced by a next farm willing to participate for the same geographical random point. A median of 3 (range 1–14) individuals were enrolled from the farms with a median of 4 LSFPs (range 1–45). Amongst the LSFPs with available data (684), 487 (71%) were farm workers or herdsmen (including several contracted wool shearers), 173 (25%) were farm/livestock owners or managers, and 24 (4%) were family members, housewives, domestic helpers or drivers. Amongst the AHCWs with available data (122), 66 (54%) were veterinarians, 37 (30%) were veterinary technicians, animal health technicians or para-veterinarians, 9 (7%) were veterinary nurses, 5 (4%) were researchers, 3 (3%) worked in game/nature conservation, and 2 (2%) indicated they practiced farming (Appendix A).

### 3.2. Seroprevalence and Factors Associated with Human Exposure

The RVFV apparent seroprevalence was 9.1% (62/685) among LSFPs and 8.0% (11/138) among AHCWs (*p* = 0.87). Adjusting for test sensitivity and specificity and for clustering, the estimated true seroprevalence was 9.1% (CI95%: 7.2–11.5%) and 8.0% (CI95%: 4.5–13.8%) among LSFPs and AHCWs, respectively.

On multivariable analysis (Table 1), adjusting for intra-farm clustering, factors associated with increased risk of RVFV seropositivity were: slaughtering animals (odds ratio (OR) = 3.9; CI95%: 1.2–12.9); preparing/consuming meat of hooved animals found dead (OR = 3.1; CI95%: 1.5–6.6); working on farm with one or more man-made dam structures for holding water (OR = 2.7; CI95%: 1.0–6.9); and injection of and collection of samples from animals (OR = 2.3; CI95%: 1.0–5.3). There was also a distinct difference in prevalence of RVFV antibody between age groups and those that had experienced one versus two or more large epidemics: the seropositivity of 30–39 age group (6.1%; CI95%: 3.1–11.7%) versus 16–29 age group (2.6%; CI95%: 1.1–6.1%), that had both experienced only one large epidemic, was not significantly different (*p* = 0.126); however, aged 40–49 (11.4%; CI95%: 6.3–20.9%) (*p* = 0.001) and aged 50–63 (18.9%; CI95%: 12.5–27.4%) (*p* < 0.001) that had both experienced two epidemics and 64 years of age and older (17.5%; CI95%: 8.4–32.9%) (*p* < 0.001) that had experienced three epidemics were all more likely to be seropositive than the youngest age group (aged 16–29). There was no significant difference between age group that experienced three versus two epidemics (*p* = 0.539). The youngest seropositive individual was a 23-year-old farm worker. 

In addition, taking measures against mosquito bites (use of skin or coil repellents) (OR: 0.52; CI95%: 0.29–0.90), and working on farms with animals used for a variety of purposes (e.g., milk ceremonial, wealth, bartering, resale, tourism) compared to one purpose breeds for meat production (OR = 0.17; CI95%: 0.03–0.93) and that has kept cattle compared to those that have not (OR = 0.35; CI95%: 0.14–0.88) were associated with lower odds of RVFV seropositivity. A lower odds of seropositivity was found in people that assisted with surgery on farm animals (OR = 0.38; CI95%: 0.15–0.98). There was no association between the presence of seropositive animals (including both vaccinated and naturally exposed animals) and the serostatus of the humans associated with that farm (univariable *p* = 0.795). 

Amongst veterinary professionals (Table 2), similar to the LSFPs, the proportion of positive samples increased markedly in subjects older than 50 years: those aged 50–63 (OR = 92.5; CI95%: 7.2–1196) and those 64 or older (OR = 167; CI95%: 7.1–3916) were much more likely to have antibodies against RVFV than subjects younger than 50 years of age. No other factors for risk of RVFV exposure were retained as significant during multivariable analysis. 

### 3.3. Estimation of RVF Cases in Farm Population

The values for the parameters defining the distributions used in the Monte Carlo simulation are given in Appendix A. The simulation estimated that the most likely number of historically RVFV infected farm workers in the study area was 3849 (95% credible interval: 2635–5374) (Appendix A).

### 3.4. Spatial Distribution of RVFV Farm Seroprevalence 

Almost 25% (44/189) of the mapped farms had at least one seropositive individual. Amongst those, percentage of seropositive individuals ranged from 15–100% on a given farm. Figure 1 shows the distribution of farms with respective seroprevalence of human RVFV and sample size. The spatial patterns of farm seroprevalence of human RVFV appeared not significantly different than random by global Moran’s *I* statistic (*z* = −0.412, *p* = 0.681). Instead Getis-Ord Gi* analysis was carried out as the spatial pattern was not consistent across the study area. Hot and cold spots of farm-level RVFV seropositivity locally within the study area were identified based on results of the Getis-Ord Gi* model of difference of observed and predicted seroprevalences, which indicated locations with causes for hot spots other than age (Figure 2). Three hot spots of human RVFV seropositivity were identified: A hotspot of high intensity was located towards western central part of the study area which is the northwestern part of the Xhariep District, encompassing Jacobsdal and Koffiefontein (Getis-Ord Gi* *p* < 0.05). 

The second hotspot was located in the northeastern part of the study area which is the middle of Lejweleputswa District, in the west of Brandfordt area (Getis-Ord Gi* *p* < 0.05). A third hotspot was found in the Nord central part, which is Boshoff area (Getis-Ord Gi* *p* < 0.05).

Three cold zones were identified in the northwestern part (Kimberley-Barkley-West), central part (Petrusburg surrounds) and eastern central part of the study area (east of Bloemfontein), of which none were significant by Getis-Ord Gi*. 

## 4. Discussion

Our seroprevalence study achieved RVFV estimates for two high-risk populations in a high-outbreak area in South Africa. Based on seroprevalence and farm population estimates we were able to provide a conservative estimate of the cumulative number of infections that had occurred in the area in order to compare it with the number of confirmed reported cases during past outbreaks in South Africa. We also obtained the first multivariable logistic regression model identifying risks for RVFV exposure in high-risk population of high-outbreak area. We were able to use it to substantiate what had been reported from outbreaks and Archer et Al. [9] study in confirmed clinical cases. Finally, we created a map to show levels and variability of farm seroprevalence and past exposure of human RVFV within the study area. 

### 4.1. Seroprevalence

The study identified a true RVFV seroprevalence of 9.1% in 685 people from primarily private livestock farms four years after the 2010–2011 RVF outbreak. Another study conducted also in the central plateau, but three years after the 1974–1975 RVF outbreak, reported a seroprevalence of 14.5% in a farming community (68 farms, 1162 participants) [16]. Although the 1978 survey included farms within our study site, it only sampled known outbreak farms and also included sites that were quite distant; this, along with the differing duration since outbreak, study design and population’s age distribution and serological tests may explain the discrepancy between the observed seroprevalences, making it difficult to determine whether there was indeed a true difference in the extent of human exposure between the two epidemics. 

Serological surveys conducted in other parts of the country that are considered low risk for RVF outbreak due to the low reported case numbers [20,30] published variable findings. A value of 10% in rural people near the eastern coast in northern KwaZulu-Natal during 1955–1958 [31], which is similar to our and McIntosh et al. estimates [16]. Surveys reported less than 1% seroprevalence in 333 Kruger National Park (KNP) personnel during 2013–2014 [32] and 0% in 64 veterinary staff and livestock workers at government diptanks in Mpumalanga [33], while we reported 8.0% among 138 veterinary professionals. The low detection compared to our study area could be explained by low virus activity or be due to high endemicity in wildlife (as in the KNP) [34], or an absence of large populations of wildlife and livestock. It is also expected that park personnel have different types of RVFV associated exposure risks as they are working with wild animals. There may also be difference in the sampled age when comparing the one survey to other surveys of veterinary professionals.

Our age group-specific results confirm the known history that no RVF epidemics occurred following the 1974–1975 outbreak until 2010–2011 within this farming community, but the increases in seroprevalence in the older age groups clearly demonstrate higher seropositivity in people that could have experienced one or more RVF epidemic. 

Inter-country comparison between published studies is complicated because the population surveyed, implementation time and scope and diagnostic assay varied by study region [35,36,37]. Similar results to our estimate of 4.3% in <40-year-olds that experienced one known epidemic in South Africa were detected on the island of Mayotte in 2011 (4.1%; 58/1413) [38], four years after RVF had emerged in humans in 2007. 

### 4.2. Risk Factors

As found in other studies [14,15,16,17,34,35,36], most of the significant associations in our study were between RVFV seropositive status and activities, habits, and behaviors involving exposure to blood or tissues of animals or their products, i.e., injecting animals or collection of clinical specimens, slaughtering, and consuming meat from an animal after it was found dead. There was only a weaker univariable association with assisting with the birth of animals compared to above-mentioned risks from the multivariable model.

Our results are similar to findings of a systematic review and meta-analysis on RVFV risk factors [39] using studies between 1989–2011, which identified slaughtering but also contact with aborted animal tissues, assisting with the birth, and skinning as being significantly associated with RVFV seropositivity. While fresh carcasses can be a source of infection with RVFV, the virus is quickly destroyed by cooking the meat [40]. The bulk of food-borne RVFV infections are probably from cross-contamination during meat preparation, which would implicate poor hygiene practices and/or meat contamination during processing of a carcass. The results of the aforementioned meta-analysis also confirmed drinking raw milk as associated with positive RVFV serology [39], but neither we nor Archer et al. in 2008–2011 [9,17] found such an association for South Africa. 

The majority of confirmed cases during the 2008–2011 RVF outbreak in South Africa had a documented history of physical contact with animals either through disposal of dead animals or aborted fetuses, or slaughtering of animals [9,17,39]. Mosquitoes, however, were believed to have had a substantial involvement in human transmission in major outbreaks in Egypt in 1977 [41] and in northern Kenya and southern Somalia in 1998 [42]. We also found some indication of the involvement of mosquitoes in human transmission, by finding that members of the farm population who took precautions against mosquito bites had a significantly lower odds of seropositivity than those that did not. 

Participants from farms with one or more dams had higher odds of seropositivity; this may indicate vector-borne exposure of livestock or humans directly because standing water provides breeding sites for mosquitoes previously implicated in RVFV transmission in South Africa (*Culex theileri*, *Aedes mcintoshi*, and *Aedes juppi*) and which readily bite people, thus suggesting that mosquito-borne human RVFV infection is possible in South Africa [16]. 

An age-dependent RVFV antibody increase has been consistent across various studies [39,43,44] and can be explained by the fact that older persons have had more opportunity to acquire viral infections and the RVFV IgG persists [45]. Using age categories, we demonstrated that seroprevalence was highly associated with the likely number of epidemics experienced during a person’s lifetime. This does not exclude the possibility that some exposure may have taken place during interepidemic periods but we know that in our study area the vast majority of exposure would have taken place during epidemics. 

### 4.3. Case Estimates

We estimated that between 2635 and 5374 historical RVFV infections occurred in people still alive on farms in the study area. Our projection suggests that the impact of RVF outbreaks on human health is likely much higher than previously documented. For example, of 2009 clinically suspected cases tested for the whole country during 2010–2011, only 278 were confirmed to be RVF cases [17]. Although numerous human infections were estimated to have occurred during the 1974–1975 epidemic, only 110 were laboratory confirmed [16]. 

Despite the observation of clinical RVF in people since 1930 [46], the first deaths were only reported in 1974–1975 in South Africa [16]. During the 2010–2011 outbreaks, 25 deaths were confirmed in South Africa [17]. The frequency of reported RVF complications and deaths has apparently increased in successive outbreaks from multiple African countries [44,47,48]. This rise may partly be due to re-assortment to virulence of live attenuated virus when infected livestock are vaccinated with live RVFV vaccine during outbreaks [48,49]. The increase may also partly reflect better recognition of RVF cases in people over time. We believe, however, that statistics for South Africa are reliable and all deaths were correctly attributed to RVF during past outbreaks. In contrast to uncomplicated cases, they would have required and likely had access to hospitalization in South Africa. 

In certain outbreaks, RVF was associated with frequent ocular morbidity [50,51]. Ocular complications have been described for the earlier epidemics of RVF [52,53,54], but it was not studied in the most recent epidemic in South Africa. Two percent is the percentage suffering from retinitis among clinical cases [7]. It cannot be directly deduced that this is the situation with sub-clinical cases as well. In view of the apparent increasing morbidity and mortality associated with RVF [47], the development of a commercial vaccine that is safe for use in humans is highly recommended to permit the immunisation of people in risk occupations [55]. 

### 4.4. Spatial Analysis

Local spatial tools found that farm RVFV seroprevalence was spatially clustered in certain areas. Hot and cold spots of farm seroprevalence of human RVFV adjusted for age were found, indicating there is a broad varied risk of past exposure within this 40,000 km^2^ area of study. Disparity in spatial-temporal patterns of RVF outbreaks was documented in Tanzania [35] and between rural villages in north-eastern Kenya [45]. However, further research by geographic analysis methods could explain environmental factors responsible for the hotspots in the identified areas. 

### 4.5. Limitations

The seroprevalence estimate was based on selected farms and the veterinary professional register within an outbreak-prone area, and is not representative of the country as a whole (particularly for rural households that do not own ruminant livestock). The study targeted at-risk occupational groups in a high-risk area for RVFV exposure in order to find sufficient seropositivity to power a risk factor analysis. Future research should also include other areas of South Africa, including those that are at low risk of outbreaks, to get a better understanding of how RVFV epidemiology varies geographically. 

Our calculations were based on information provided by the participants and may have been affected by recall bias and false self-reporting, e.g., it is much easier for a person to recall whether he/she had contact with animals than a mosquito bite exposure. Despite the small number of people without direct animal contact or working with game, we were able to draw some conclusions about factors associated with RVFV seropositivity which should be interpreted with caution. We used a simple simulation method to estimate cumulative numbers of infections that had historically occurred in the area. People exposed to more outbreaks may be under-represented in our sample because they might have died, and this may result in our estimates being conservative. Comparison with other areas will be limited with results obtained. We were not able to satisfy assumptions underlying other methods to calculate more informative estimates e.g., force of infection rather than a total number because estimates would vary a lot because of epidemic and occupational nature of RVFV infection amongst other. In addition, reliable population data for the study area during previous outbreaks are unavailable. Non-random sampling within farms may have affected the spatial analysis, which is dependent on accuracy of the farm-level estimates of seropositivity. If seropositive individuals moved between farms this could also have biased the spatial analysis. Geographic data were not available to analyse geographic distances between hotspots and other areas of the study.

## 5. Conclusions

Our study showed that one in ten individuals in at-risk occupations was seropositive in a region with three previously documented, large RVFV outbreaks. The study highlighted the increase in seroprevalence in people that had experienced more outbreaks. While seropositive status should provide long-lived immunity in an individual, the level identified here is not adequate for herd immunity. For that reason, prevention efforts should in the first instance be directed toward ensuring herd immunity is reached in livestock through vaccination on these farms to prevent the occurrence of outbreaks and subsequent zoonotic exposures in humans. As a second line of defense, our research identified exposure factors which could be targeted for reducing overall infections in populations during future outbreaks, including safe handling and animal slaughtering practices and wearing adequate protective equipment during injection, sample collection, slaughtering and processing of infected meat for consumption and precautions against mosquito bites. 

The distribution of RVFV seropositivity was broad but patchy in the central plateau of South Africa. Focusing RVF control measures during epidemic and inter-epidemic periods toward zones at high risk for RVF at livestock-owning household- and neighborhood-level may result in a more cost-effective approach of RVF management and prevention of spread. Assessing the relationship between RVF clusters and environmental factors could further contribute to interventions for reducing transmission of RVFV. 

Research in understanding mosquito ecology, the most important way RVFV is spread amongst animals, is a prerequisite in prevention and control measures for RVF. In addition is monitoring to understand immunity in animals to protect against infection in these and humans from zoonotic exposure. By implementing collaborative cross-species RVFV surveys in livestock, wildlife, humans, and mosquito vectors together with climate and vegetation studies according to a one-health approach we had a much better chance to improve our understanding of RVFV ecology and epidemiology in South Africa instead of one-sided self-standing studies. Research by multidisciplinary team improved interdisciplinary communication and knowledge transfer which can lay foundation for continued collaboration at the control phase. At-risk community received awareness and education from professionals with a wide range of expertise in this way contributing to more effective joint disease control. Efforts are also shared in logistic services, the development and evaluation of diagnostic methods animal vaccines and larviciding measures, for example, that benefits early detection and prevention of loss of livestock and morbidity and death in humans in more cost-effective way. Associated risks identified in both animals and humans can be prevented and controlled in one way which should be more cost-effective. RVF forecasting and climatic models a frontline control method requires input from all disciplines to improve accuracy for local setting. 

## Figures and Tables

**Figure 1 viruses-11-00140-f001:**
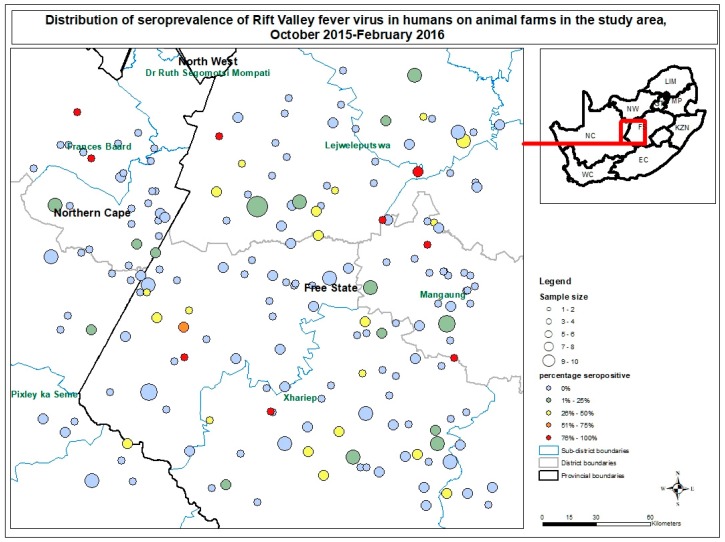
Distribution of anti–Rift Valley fever virus antibodies in 462 humans on 189 ruminant livestock and game farms in the study area in central South Africa, sampled during 2015–2016.

**Figure 2 viruses-11-00140-f002:**
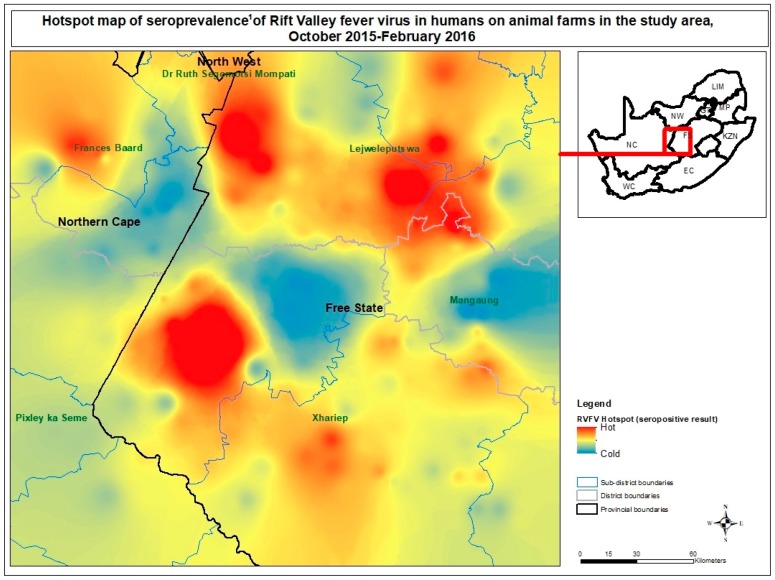
Hotspot map of anti–Rift Valley fever virus antibody prevalence, adjusted for age, ^1^ in 462 humans on 189 ruminant livestock and game farms in the study area in central South Africa, sampled during 2015–2016. ^1^ Difference of RVFV observed seroprevalence and predicted seroprevalence (by logistic regression model including age) by farm.

**Table 1 viruses-11-00140-t001:** Univariable and multivariable logistic regression analysis of potential risk factors for Rift Valley fever virus (RVFV) seropositivity in people working on farms in the study area of South Africa during 2015–2016.

Variables ^1^	RVFV Seropositive n/N (%)	Univariable Analysis		Multivariable Analysis ^2^	
		Odds Ratio (CI95%)	*p*-Value (<0.2)	Odds Ratio (CI95%)	*p*-Value (<0.05)
**Demographic characteristics**					
Age (years)					
16–29	5/196 (2.6%)	1 (base)	-	1 (base)	-
30–39	12/196 (6.1%)	2.7 (0.8–8.7)	0.100	2.64 (0.76–9.18)	0.126
40–49	13/114 (11.4%)	5.9 (1.9–18.2)	0.002	6.91 (2.15–22.2)	0.001
50–63	20/106 (18.9%)	12.6 (4.3–37.0)	<0.001	12.9 (4.15–40.0)	<0.001
≥64	7/40 (17.5%)	16.9 (4.3–66.0)	<0.001	25.6 (5.50–119)	<0.001
Working on farm (years)				Eliminated	
≤5	16/299 (5.4%)	1 (base)	-		
6–10	11/108 (10.2%)	2.1 (1.0–4.5)	0.53		
11–20	14/128 (10.9%)	2.5 (1.2–5.3)	0.019		
21–30	8/64 (12.5%)	3.1 (1.2–7.9)	0.016		
31–40	5/37 (13.5%)	4.0 (1.4–11.8)	0.012		
>40	8/48 (16.7%)	5.3 (2.0–13.6)	0.001		
Working with animals (years)				Eliminated	
≤5	10/240 (4.2%)	1 (base)	-		
6–10	8/110 (7.3%)	1.9 (0.8–4.7)	0.166		
11–20	15/145 (10.3%)	3.0 (1.3–7.3)	0.014		
21–30	15/85 (17.6%)	6.1 (2.5–15.2)	<0.001		
31–40	6/51 (11.8%)	4.5 (1.4–14.8)	0.014		
>40	7/48 (14.6%)	6.3 (2.1–18.6)	0.001		
Job					
Farm worker/herdsman	46/487 (9.5%)	1 (base)	-	1 (base)	-
Farm/livestock owner/manager	15/173 (8.7%)	0.9 (0.5–1.7)	0.758	1.26 (0.42–3.80)	0.678
Family, domestic worker, driver	1/24 (4.2%)	0.4 (0.05–3.3)	0.403	0.65 (0.06–6.72)	0.718
**Activities in past**					
Cleaning equipment				Eliminated	
Yes	53/547 (9.7%)	1.5 (0.8–2.9)	0.176		
No	9/137 (6.6%)	1 (base)	-		
Injection and collection of samples from animals					
Yes	53/500 (10.6%)	2.4 (1.2–4.7)	0.014	2.33 (1.03–5.30)	0.043
No	9/184 (4.9%)	1 (base)	-	1 (base)	-
Assisting with birth of animal				Eliminated	
Yes	57/577 (9.9%)	2.1 (0.7–6.3)	0.172		
No	5/107 (4.7%)	1 (base)	-		
Assisting with surgery					
Yes	8/136 (6.0%)	0.5 (0.3–1.1)	0.097	0.38 (0.15–0.98)	0.046
No	54/548 (9.9%)	1 (base)	-	1 (base)	-
Slaughtering of animals					
Yes	58/563 (10.3%)	3.3 (1.3–8.7)	0.014	3.93 (1.20–12.88)	0.024
No	4/121 (3.3%)	1 (base)	-	1 (base)	-
Burying dead animals				Eliminated	
Yes	45/422 (10.7%)	1.7 (0.9–3.0)	0.079		
No	17/262 (6.5%)	1 (base)	-		
Eating hooved animal found dead					
Yes	42/342 (12.3%)	3.4 (1.8–6.7)	<0.001	3.14 (1.49–6.61)	0.003
No	20/342 (5.8%)	1 (base)	-	1 (base)	-
Measures against mosquito bites					
Yes	22/305 (7.2%)	0.7 (0.4–1.1)	0.114	0.52 (0.29–0.90)	0.021
No	40/379 (10.6%)	1 (base)	-	1 (base)	-
Working on farm with primarily domestic or wild animals				Eliminated	
Wild	3/16 (18.8%)	2.3 (1.2–4.4)	0.011		
Domestic	59/669 (8.8%)	1 (base)	-		
Working on farm with private or communal land use ^3^					
Communal	3/44 (6.8%)	0.8 (0.3–2.4)	0.656	0.75 (0.09–5.93)	0.784
Private	59/641 (9.2%)	1 (base)	-	1 (base)	-
Working on farm that kept cattle					
Yes	47/581 (8.1%)	0.5 (0.2–1.0)	0.059	0.35 (0.14–0.88)	0.025
No	15/104 (14.4%)	1 (base)	-	1 (base)	-
Manmade dam(s) on farm					
Yes	54/532 (10.2%)	2.5 (1.1–5.7)	0.032	2.68 (1.04–6.89)	0.041
No	6/139 (4.3%)	1 (base)	-	1 (base)	-
New animals are quarantined				Eliminated	
Yes	7/142 (4.9%)	0.4 (0.2–1.1)	0.069		
No	55/543 (10.1%)	1 (base)	-		
Main purpose of farming ^3^					
Meat	40/367 (10.9%)	1 (base)	-	1 (base)	-
Dairy	1/24 (4.2%)	0.3 (0.03–3.6)	0.373	1.00 (0.13–7.56)	0.999
Meat-wool	15/194 (7.7%)	0.7 (0.4–1.2)	0.186	0.67 (0.34–1.29)	0.225
Other (milk, bartering, wealth, ceremonial, resale, tourism)	6/100 (6.0%)	0.5 (0.2–1.3)	0.169	0.17 (0.03–0.93)	0.041
Animals are slaughtered on farm				Eliminated	
Yes	47/443 (10.6%)	1.8 (0.8–4.2)	0.158		
No	12/198 (6.1%)	1 (base)	-		
Animals vaccinated against RVFV in the past				Eliminated	
Yes	41/357 (11.5%)	1.9 (1.1–3.3)	0.027		
No	19/300 (6.3%)	1 (base)	-		
RVF on farm in past incl. participants working there 4 years or more only				Not included	
Yes	19/149 (12.8%)	1.3 (0.6–2.7)	0.498	Checked for confounding	
No	28/272 (10.3%)	1 (base)	-		
Drinking milk				Not included	
Pasteurised/boiled	19/233 (8.2%)	1 (base)			
On occasion raw	5/56 (8.9%)	1.2 (0.4–3.6)	0.762		
Raw	27/292 (9.2%)	1.1 (0.5–2.2)	0.817		

^1^ The variables with univariable *p*-value < 0.2 were included in the multivariable analysis. The variables with multivariable *p*-value < 0.05 were kept in the multivariable model. ^2^ Eliminated means that the variable was first included in the model and then it was omitted due to the fact that its *p*-value in the model was ≥ 0.05. ^3^ Most communal farmers farm with mixed purpose while private land-owned farms usually specialize in one or dual production motive. When landownership variable was eliminated from the model the odds for seropositivity in mixed production purpose gave unrealistic results and for this reason landownership was retained as potential confounder in the model despite its statistical insignificance. Sex was not significant by univariable analysis (*p* ≥ 0.2). Sex was not considered as potential confounder and most farm workers were male.

**Table 2 viruses-11-00140-t002:** Univariable and multivariable logistic regression analyses for potential risk factors for RVFV seropositivity in veterinarians and associated professions (AHCWs) in the study area of South Africa during 2015–2016.

Variables ^1^	RVFV Seropositive n/N (%)	Univariable Analysis		Multivariable Analysis ^2^	
		Odds Ratio (CI95%)	*p*-Value (<0.2)	Odds Ratio (CI95%)	*p*-Value (<0.05)
**Demographic characteristics**					
**Age (years)**					
16–49	1/98 (1.0%)	1 (base)	-	1 (base)	-
50–63	6/22 (27.3%)	92.5 (7.2–1196)	0.001	92.5 (7.16–1196)	0.001
≥64	2/7 (28.6%)	166 (7.2–3916)	0.002	167 (7.08–3916)	0.002
Working as AHCW (years)				Eliminated	
≤40	1/85 (1.2%)	1 (base)	-		
41–50	5/22 (22.7%)	62.4 (2.07–1886)	0.018		
≥51	4/15 (26.7%)	65.4 (3.9–1110)	0.004		
Job description					
Animal health tech	2/37 (5.4%)	1 (base)	-	1 (base)	-
Veterinarian	6/66 (9.1%)	1.75 (0.40–7.57)	0.450	0.11 (0.01–1.72)	0.116
Other (incl. vet nurse, researcher, wildlife capturers) ^3^	2/19 (10.5%)	2.06 (0.28–15.03)	0.473	0.74 (0.05–11.01)	0.825
**Activities in past**					
Cleaning waste				Eliminated	
Yes	9/83 (10.8%)	3.4 (0.7–16.6)	0.137		
No	2/55 (3.6%)	1 (base)	-		
Working with hoofed animals				Eliminated	
<1 h	3/67 (4.5%)	1 (base)	-		
Half day	3/37 (8.1%)	1.7 (0.3–9.7)	0.548		
Whole day	5/34 (14.7%)	3.6 (0.7–17.3)	0.110		
Contact with RVF positive animals in the past				Eliminated	
Yes	9/82 (11.0%)	5.1 (0.7–38.6)	0.112		
No	1/40 (2.5%)	1 (base)	-		
Taking measures against mosquito bites				Eliminated	
Yes	4/84 (4.8%)	0.4 (0.1–1.6)	0.193		
No	7/54 (13.0%)	1 (base)	-		
Drinking milk				Eliminated	
Yes	8/125 (6.4%)	0.18 (0.03–0.98)	0.047		
No	3/13 (23.1%)	1 (base)	-		
**Conditions**					
On chronic medication				Eliminated	
Yes	5/43 (11.6%)	2.6 (0.7–9.8)	0.170		
No	6/95 (6.3%)	1 (base)			
Chronic liver disease				Eliminated	
Yes	1/2 (50%)	11.8 (0.6–229)	0.102		
No	10/136 (7.4%)	1 (base)	-		

^1^ The variables with univariable *p*-value < 0.2 were included in the multivariable analysis. The variables with multivariable *p*-value < 0.05 were kept in the multivariable model. ^2^ Eliminated means that the variable was first included in the model and then it was omitted due to the fact that its *p*-value in the model was ≥ 0.05. ^3^ Sex was not significant by univariable analysis (*p* ≥ 0.2). Sex was not considered as potential confounder and sexes were equally distributed in veterinary professionals.

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
