# Peer review of "Rift Valley Fever Virus Exposure amongst Farmers, Farm Workers, and Veterinary Professionals in Central South Africa"

_viruses, 2019, doi:10.3390/v11020140_

Round 1

Reviewer 1 Report

The study described in the manuscript by Msimang et al. is aimed at quantifying the prevalence of previous infection by Rift Valley Fever (RVF) virus among farm population in    central South-Africa. Specifically, it aims to quantify RVF seroprevalence among farmers, farm workers and veterinary personnel in this area, to determine risk factors for infection by RVFV, to identify hot-spots and to estimate the true number of infected people in this area during previous outbreak.

Though this study does not provide new findings regarding the epidemiology of RVF, it's aims are of high importance and it is nicely performed. Thus, besides its main contribution to the local community in the studied area I believe that it provides a nice contribution to the knowledge regarding this important disease. 

The methodology used by the authors is valid for achieving the two first aims. As I will outline later I think that the fourth aim (which is the novelest one) is achievable too with some recommended additions to the analysis.  Another adjustment should be also maid for achieving the third aim.

Following are specific comments and suggestions to improve the article:

Lines 32-5: The sentence is too long and not clear. I suggest rephrasing and dividing to two sentences.

Line 35: The sentence starts with "And…". Please rephrase.

Lines 38-40: Please add the central estimate to the 95% credible interval.

Lines 49-57: I suggest adding some short information on the causing virus and on the possible routes of its transmission.

line 63: "expansive" or "intensive"?

line 70: add "in" before "improvement".

Lines 86-95: I recommend adding a table of demographic and age data on the sampled farmers and veterinary professionals. I know that these data are already incorporated in tables 1 and 2. However, these tables are designed to show the results and it is difficult to compare between these groups in that way. Average and distribution of the farm size should be provided as well.

Lines 101-3: Please provide data on refusal proportion among farm owners, among target individuals in the selected farms and among veterinary personnel.

Line 106-10: Please add the questionnaire (in English) to the supplementary material.

Lines 126-8: This is not an appropriate statistical language. Please define what model did you use (did you use a generalized linear model with a mixed effect design? What is the link function? Is it a logistic model?)

Line 132-4: So here you used a logistic regression. But again, for the people who do not use stata but other statistical packages. What is this svy procedure?

Line 134-5: You interchangeably use the term multivariable and univariate, while you mean multivariable and univariable (as all the analysis you perform, (whether univariable or multivariable) is univariate). This is true for tables 1 and 2, as well, and in other places in the manuscript.

Lines 136-8: Please detail all the variables that were inserted in the model regardless of their significance. And please explain, why for example ownership of the land is more important as a potential confounder than other variable, so that it has to be included regardless of its significance. Please provide such an explanation for all other variables included in such a way.

Lines 139-160: I do not fully agree with the rationale behind the calculation of the affected population. Population may have changed during the long time from the first outbreak to the conduction of the serological survey. The percentage of RVF sero-positive people in a survey, also does not represent accurately the number of people affected in each outbreak, as this depends on the age distribution of the sampled population. People that lived during the first outbreak are underrepresented in the population as some of them might have died just due to old age. As an alternative I suggest revising the analysis method. By using the seropositivity in each age distribution it can be deduced retrospectively what percentage of the susceptible population was affected in each outbreak (for more detail on this methodology please look at Cohen, D.I., Davidovici, B.B., Smetana, Z. et al. Infection (2006) 34: 208. https://doi.org/10.1007/s15010-006-6604-4). Then, the population at risk during each outbreak should be estimated, as it is obvious that the population in that area during the outbreak which occurred during the outbreaks of 1950-1 does not resemble the population leaving in that area 70 years later. Thus an estimation of the population size at each outbreak should be included in the analysis along with a measure of the uncertainty of these estimates. These populations should be multiplied by the estimated percentage of exposure calculated by the methodology I suggested and added together to provide a better estimate of the total population affected during these outbreaks.

Lines 161-180: Clusters found in this analysis can be a result of confounders. For example, it can be that such a cluster is due to random clustering of older people in a certain location. This should be dealt with as it does not have a biological interpretation but rather an interpretation related to sampling. I do not know the method to control for this with the calculation of the local Moran's I estimate. However, using the scan statistic (see https://www.satscan.org/) clusters can be calculated after controlling for age. An alternative is to calculate the local Moran's I statistic of the residuals from a model of the association of age with seropositivity.

Tables 1 and 2: Univariable and not univariate.

What the meaning of 'eliminated' is? I assume you mean that the variable was first included in the model and then it was omitted due to the fact that its p-value in the model was higher than 0.05. Pleas add a footnote explaining this.

Is there a reason to the fact that sex was not included in the analysis?

Line 283-301: Another possible explanation is a difference in the sampled age. This is also true when comparing the veterinarian survey to other surveys of veterinarians.

Lines 347-70: As I outlined earlier the entire calculation of the number of affected cases is problematic. Therefore, this section should be rewritten after new calculation of the affected population.

Lines 366-8: I think that the conclusion regarding the number of people suffering from retinitis is problematic. 2% is the percentage suffering from retinitis among clinical cases. It cannot be directly deduced that this is the situation with sub-clinical cases as well.

Line 373-77: This part of the discussion is a little shallow. Please discuss if there are any apparent geographical distances between the hot spots and the other study areas (e.g. altitude, proximity to water bodies). Please refer also to the mosquito spp. abundant in this area and their habitats. I should mention again here that the clusters should be recalculated after controlling for major confounder like age for example.

Author Response

Response to Reviewer 1 Comments

Point1: Lines 32-5: The sentence is too long and not clear. I suggest rephrasing and dividing to two sentences.

Response 1: Sentence was rephrased.

Point 2: Line 35: The sentence starts with "And…". Please rephrase.

Response 2: Sentence was rephrased.

Point 3: Lines 38-40: Please add the central estimate to the 95% credible interval.

Response 3: Central estimate was added.

Point 4: Lines 49-57: I suggest adding some short information on the causing virus and on the possible routes of its transmission.

Response 4: Information was added about Rift Valley fever virus and its routes of transmission.

Point 5: line 63: "expansive" or "intensive"?

Response 5: Changed in text

Point 6: line 70: add "in" before "improvement".

Response 6:  Changed in text

Point 7: Lines 86-95: I recommend adding a table of demographic and age data on the sampled farmers and veterinary professionals. I know that these data are already incorporated in tables 1 and 2. However, these tables are designed to show the results and it is difficult to compare between these groups in that way. Average and distribution of the farm size should be provided as well.

Response 7: In the text, I added proportion of sex of participants by occupational group. I also added median farm size to the text. Supplementary material now has a table with these data.

Point 8: Lines 101-3: Please provide data on refusal proportion among farm owners, among target individuals in the selected farms and among veterinary personnel.

Response 8: Information added to text

Point 9: Line 106-10: Please add the questionnaire (in English) to the supplementary material.

Response 9: Participant survey added to Supplementary material

Point10: Lines 126-8: This is not an appropriate statistical language. Please define what model did you use (did you use a generalized linear model with a mixed effect design? What is the link function? Is it a logistic model?)

Response 10: Changed in text.

Now reads as: Line 140: “We estimated proportion of seroprevalence and constructed confidence intervals adjusted for clustering at farm level by using the linearized variance estimator based on a first-order Taylor series linear approximation (Wolter 2007- Introduction to variance estimation). 

Analysis was done using Stata 13 (StataCorp, College Station, TX, USA) with adjustment for data that were collected using a survey sampling design and clustering using the svy-set command that specified the farm identifier as the primary sampling unit (cluster) variable and svy-prefix for estimation and risk factor analysis commands.”

Point11: Line 132-4: So here you used a logistic regression. But again, for the people who do not use stata but other statistical packages. What is this svy procedure?

Response 11: Variance estimates to account for clustering as mentioned above.

Point12: Line 134-5: You interchangeably use the term multivariable and univariate, while you mean multivariable and univariable (as all the analysis you perform, (whether univariable or multivariable) is univariate). This is true for tables 1 and 2, as well, and in other places in the manuscript.

Response 12: Changed to univariable in text.

Point13: Lines 136-8: Please detail all the variables that were inserted in the model regardless of their significance. And please explain, why for example ownership of the land is more important as a potential confounder than other variable, so that it has to be included regardless of its significance. Please provide such an explanation for all other variables included in such a way.

Response 13: These lines describe methods. Which variables which are included are a result of the univariable analysis and therefore cannot be listed under methods. The variables inserted in the multivariable model are given in detail in results in table 1. The reason for retention of landownership is now explained under results Table 1.

Point14: Lines 139-160: I do not fully agree with the rationale behind the calculation of the affected population. Population may have changed during the long time from the first outbreak to the conduction of the serological survey. The percentage of RVF sero-positive people in a survey, also does not represent accurately the number of people affected in each outbreak, as this depends on the age distribution of the sampled population. People that lived during the first outbreak are underrepresented in the population as some of them might have died just due to old age. As an alternative I suggest revising the analysis method. By using the seropositivity in each age distribution it can be deduced retrospectively what percentage of the susceptible population was affected in each outbreak (for more detail on this methodology please look at Cohen, D.I., Davidovici, B.B., Smetana, Z. et al. Infection (2006) 34: 208. https://doi.org/10.1007/s15010-006-6604-4). Then, the population at risk during each outbreak should be estimated, as it is obvious that the population in that area during the outbreak which occurred during the outbreaks of 1950-1 does not resemble the population leaving in that area 70 years later. Thus an estimation of the population size at each outbreak should be included in the analysis along with a measure of the uncertainty of these estimates. These populations should be multiplied by the estimated percentage of exposure calculated by the methodology I suggested and added together to provide a better estimate of the total population affected during these outbreaks.

Response 14: Cohen et al. (2006) used age-specific seroprevalence to estimate force of infection (FOI) of varicella zoster virus, an endemic pathogen circulating at high levels, using catalytic models, allowing FOI to vary by age category. In theory, if one knew the population and age distribution during various time intervals, one could calculate the number of infections that occurred during each. However, an important limitation of this method, which makes it unsuitable for use with our data (epidemic rather than endemic RVFV infection), is that such catalytic models assume that FOI remains constant over time within each age group. Alternatively, FOI can be allowed to vary over time but be constant between age categories. Either way, such assumptions would not hold for our dataset, where FOI would have shown extreme variation, being zero (or almost zero) between epidemics. Even during epidemics, in our setting FOI would also have varied greatly between geographic locations and between age groups due to different ecological and occupational exposure factors. Thus, catalytic models, to our knowledge, would be unsuitable for our purposes. In addition, we are not able to obtain reliable population estimates, including age distributions, for our study area during previous epidemics. We agree that people exposed to more outbreaks may be under-represented in our sample because they might have died, and this may result in our estimates being conservative. However, we think this is preferable to using a method that requires unrealistic assumptions and where the direction of bias may be difficult to predict.

We therefore prefer to use our relatively simple simulation method which, although imperfect, makes use of the available information to provide a conservative and more easily defensible estimate of the number of infections. Additional discussion of the limitations of our method has been added [see if this is necessary].

Point15: Lines 161-180: Clusters found in this analysis can be a result of confounders. For example, it can be that such a cluster is due to random clustering of older people in a certain location. This should be dealt with as it does not have a biological interpretation but rather an interpretation related to sampling. I do not know the method to control for this with the calculation of the local Moran's I estimate. However, using the scan statistic (see https://www.satscan.org/) clusters can be calculated after controlling for age. An alternative is to calculate the local Moran's I statistic of the residuals from a model of the association of age with seropositivity.

Point 15: We predicted mean RVFV seroprevalence of people that had worked/lived at that farm for 4 years or longer, by fitting a logistic regression model to age data in Stata. We subtracted predicted from actual mean farm seroprevalence of human RVFV and ran a hotspot analysis on the difference in seroprevalences. This was done to verify whether external factors other than age caused hot and cold spots. We used the hotspot analysis to evaluate the clusters. Changes are made to text in methods and results and discussion and maps.

Point16: Tables 1 and 2: Univariable and not univariate.

Response 16: Changed to univariable.

Point17: What the meaning of 'eliminated' is? I assume you mean that the variable was first included in the model and then it was omitted due to the fact that its p-value in the model was higher than 0.05. Pleas add a footnote explaining this.

Response 17: Footnote explaining “eliminated” in footnote of table 1 and 2 added.

Point18: Is there a reason to the fact that sex was not included in the analysis?

Response 18: Sex was not significant by univariable analysis (p>0.2). Sex was not considered as potential confounder and most farm workers were male and sexes were equal in veterinary professionals.

Point19: Line 283-301: Another possible explanation is a difference in the sampled age. This is also true when comparing the veterinarian survey to other surveys of veterinarians.

Response 19: Added to text

Point20: Lines 347-70: As I outlined earlier the entire calculation of the number of affected cases is problematic. Therefore, this section should be rewritten after new calculation of the affected population.

Response 20: I refer to response 15

Point21: Lines 366-8: I think that the conclusion regarding the number of people suffering from retinitis is problematic. 2% is the percentage suffering from retinitis among clinical cases. It cannot be directly deduced that this is the situation with sub-clinical cases as well.

Response 21: I removed this conclusion about retinitis from the text.

Point22: Line 373-77: This part of the discussion is a little shallow. Please discuss if there are any apparent geographical distances between the hot spots and the other study areas (e.g. altitude, proximity to water bodies). Please refer also to the mosquito spp. abundant in this area and their habitats. I should mention again here that the clusters should be recalculated after controlling for major confounder like age for example.

Response 22:

We merely wanted to show levels and variability of farm seroprevalence and past exposure of human RVFV within study area. We did not collect and analyse geographical data and studying geographical distances between hotspots and colder study areas was not part of our study design. Despite that, the study was conducted in the framework of a larger one-health project with human, animal as well as vector surveillance together with vegetation and climate studies, which will analyse geographical factors amongst other in the study area.

Reviewer 2 Report

This is a well written and referenced examination of the seroprevalence and associated risk factors of human exposure to RVFV in a logically selected study population which has had significant opportunity for prior exposure to the virus. The recommendations and findings in this study reinforce findings of prior studies and inform future public health practices as regards protecting human populations from zoonotic transmission of RVFV. As a RVFV researcher I appreciated the opportunity to review this manuscript. This work is highly relevant since it targeted an area in S Africa where multiple well documented outbreaks occurred prior and are likely to occur again. Yet this study doesn’t overstate the relevance of its findings given its limitations. 

My recommendations are all minor. However, there is one caveat, my expertise doesn’t extend to an ability to critique and review in detail the mathematical modeling and statistics used in this work and I recommend that this be done by another reviewer.  

Minor comments:

There are a few places where extra spaces are included or missing from the text.

Intro:

This could be improved by providing a bit more on the type of virus RVFV is including its taxonomic assignment.

Also mention that humans are often he sentinel species in a RVFV outbreak would be useful. Interestingly, this was not the case in the small 2018 outbreak mentioned but has typically been so prior. This reinforces the importance of the need for herd immunity in susceptible ruminants as well as at risk humans, if possible.

M&M:

-Line 80: “;” in the ethics statement ahead of “SA Provincial” s grammatical odd, please adjust.

-Line 114: correct spelling to “venipuncture” 

Discussion:

-An introduction paragraph to the discussion section identifying overall key findings would be helpful.

-Line 299: Why would high endemicity in wildlife drive low seroprevalence in humans save if by it being being endemic the viral shedding is markedly decreased and therefore human exposure minimized?

-Line 305: “epidemic” should be epidemics

-Line 306: “the” population and in next line no comma needed after “assay”

-Line 318: While I understand what is meant by there is only a “univariate association” not all readers necessarily will and perhaps clarity regarding this beating a weaker association than the aforementioned ones would help the reader.

-Line 319: “factors”not “factor”

-Line 326: “associated” not “association” ? And it would appear that this is in reference to the prior meta-analysis although that is not clear in this sentence as lacks a reference or other indicator. Please rephrase.

-Line 343: “persistence “is for several years to life long

-Line 358: “multiple African countries”

-Line 360: “outbreaks” 

-Line 361: “are reliable”

-Line 369: more accurately the morbidity and mortality is associated with the disease RVF not the virus RVFV

-Line 407: please elaborate what “special precautions” might be

Author Response

Response to Reviewer 2 Comments

Point 1: Intro:

This could be improved by providing a bit more on the type of virus RVFV is including its taxonomic assignment.

Also mention that humans are often the sentinel species in a RVFV outbreak would be useful. Interestingly, this was not the case in the small 2018 outbreak mentioned but has typically been so prior. This reinforces the importance of the need for herd immunity in susceptible ruminants as well as at risk humans, if possible.

Response 1: Has been added to the text

M&M:

Point 2: Line 80: “;” in the ethics statement ahead of “SA Provincial” is grammatical odd, please adjust.

-Line 114: correct spelling to “venipuncture” 

Response 2: Has been adjusted in text.

Discussion:

Point 3: An introduction paragraph to the discussion section identifying overall key findings would be helpful.

Now reads as: “Our seroprevalence study achieved RVFV estimates for two high risk populations in a high-outbreak area in South Africa. Based on seroprevalence and farm population estimates we were able to provide a conservative estimate of the cumulative number of infections that had occurred in the area in order to compare it with the number of confirmed reported cases during past outbreaks in South Africa. We also obtained first multivariable logistic regression model identifying risks for RVFV exposure in high risk population of high-outbreak area. We were able to use it to substantiate what had been reported from outbreaks and Archer et Al. study in confirmed clinical cases. Finally we created a map to show levels and variability of farm seroprevalence and past exposure within the study area.”

Point 4: Line 299: Why would high endemicity in wildlife drive low seroprevalence in humans save if by it being being endemic the viral shedding is markedly decreased and therefore human exposure minimized?

Response 4: That is correct. In case of endemicity, there is very low number of viraemic animals and therefore lower risk of exposure to humans.

Point 5:

- Line 305: “epidemic” should be epidemics

-Line 306: “the” population and in next line no comma needed after “assay”

-Line 318: While I understand what is meant by there is only a “univariate association” not all readers necessarily will and perhaps clarity regarding this beating a weaker association than the aforementioned ones would help the reader.

-Line 319: “factors”not “factor”

-Line 326: “associated” not “association” ? And it would appear that this is in reference to the prior meta-analysis although that is not clear in this sentence as lacks a reference or other indicator. Please rephrase.

-Line 343: “persistence “is for several years to life long

-Line 358: “multiple African countries”

-Line 360: “outbreaks” 

-Line 361: “are reliable”

-Line 369: more accurately the morbidity and mortality is associated with the disease RVF not the virus RVFV

-Line 407: please elaborate what “special precautions” might be

Response 5: All points been adjusted in text.

Reviewer 3 Report

The manuscript by Msimang et al investigated the prevalence of RVF virus among the high risk group of livestock and wildlife farmers and animal health care workers in central South Africa, a region that has experienced multiple outbreaks of RVF. The study was conducted in 2015-2016, while the last reported outbreak in the region was in 2010-2011. The study reported prevalence of 8-9% among the two groups. While the findings are important and merit publication, the manuscript has some weaknesses that should be addressed.

1. The data of estimation of the number of RVF cases in the region is meaningless without providing a denominator in order to express this as a rate. These should be removed from the paper.

2. These prevalence data on this high risk group would have been more informative if it was compared to a low-risk population. Right now, it is hard to determine if this group is any higher risk of RVF infections than the rest of the population from the study. At the minimum, the authors should compare these findings with similar studies on low risk populations in South Africa or elsewhere.

3. The reported increase in RVF seropositivity with age is not just a function of RVF epidemics but also endemic exposure (without documented outbreaks) as has been demonstrated in many studies including the ones listed below. The authors should add this possibility into the discussion.

              Nyakarahuka L, de St Maurice A, Purpura L, Ervin E, Balinandi S, Tumusiime A, et al. Prevalence and risk factors of Rift Valley fever in humans and animals from Kabale district in Southwestern Uganda, 2016. PLoS Negl Trop Dis 2018: 12:e0006412. doi: 10.1371/journal.pntd.0006412.

              Gudo ES, Pinto G, Weyer J, le Roux C, Mandlaze A, José AF, et alSerological evidence of rift valley fever virus among acute febrile patients in Southern Mozambique during and after the 2013 heavy rainfall and flooding: implication for the management of febrile illness. Virol J 2016:13:96. doi: 10.1186/s12985-016-0542-2.

              Njenga MK & Bett B. Rift Valley fever virus: Where and How it is maintained during inter-epidemic periods. Curr Clin Microbiol Reprts. 2018: 1-7 10.1007/s40588-018-0110-1

Author Response

Response to Reviewer 3 Comments

Point 1: The data of estimation of the number of RVF cases in the region is meaningless without providing a denominator in order to express this as a rate. These should be removed from the paper.

Response 1: Our objective was not to estimate the rate of infection but merely to provide a conservative estimate of the number of infections that had occurred in the area in order to compare it with the number of confirmed reported cases during past outbreaks in South Africa. Calculation of rates is problematic due to high variation in time and other factors.  We know that in this geographical high-outbreak area of South Africa, the greater part are epidemic infections rather than from inter-epidemic acquisition. Even if we were to estimate it only for epidemics, RVFV infections merely involve occupational groups that have contact with hooved animals and we do not have reliable population estimates, including age distributions, in our study area during previous epidemics.

Point 2. These prevalence data on this high risk group would have been more informative if it was compared to a low-risk population. Right now, it is hard to determine if this group is any higher risk of RVF infections than the rest of the population from the study. At the minimum, the authors should compare these findings with similar studies on low risk populations in South Africa or elsewhere.

Response 2: The aim of the study was to estimate the seroprevalence in a high risk population and not comparing between populations.  In the discussion, comparison was however made with results from other studies from South Africa and other countries (Lines 248-311).

Point 3. The reported increase in RVF seropositivity with age is not just a function of RVF epidemics but also endemic exposure (without documented outbreaks) as has been demonstrated in many studies including the ones listed below. The authors should add this possibility into the discussion.

              Nyakarahuka L, de St Maurice A, Purpura L, Ervin E, Balinandi S, Tumusiime A, et al. Prevalence and risk factors of Rift Valley fever in humans and animals from Kabale district in Southwestern Uganda, 2016. PLoS Negl Trop Dis 2018: 12:e0006412. doi: 10.1371/journal.pntd.0006412.

              Gudo ESPinto GWeyer Jle Roux CMandlaze AJosé AF, et al.  Serological evidence of rift valley fever virus among acute febrile patients in Southern Mozambique during and after the 2013 heavy rainfall and flooding: implication for the management of febrile illness. Virol J 2016:13:96. doi: 10.1186/s12985-016-0542-2.

              Njenga MK & Bett B. Rift Valley fever virus: Where and How it is maintained during inter-epidemic periods. Curr Clin Microbiol Reprts. 2018: 1-7 10.1007/s40588-018-0110-1

Response 3: We have added endemic exposure (without documented outbreaks) as a possibility to the text in discussion.

It now reads as: “This does not exclude the possibility that some exposure may have taken place during interepidemic periods but we know that in our study area the vast majority of exposure would have taken place during epidemics.”

Round 2

Reviewer 1 Report

none